

# Machine Learning Detection of Dust Impact Signals Observed by The Solar Orbiter

Andreas Kvammen[1], Kristoffer Wickstrøm[1], Samuel Kociscak[1], Jakub Vaverka[2], Libor Nouzak[2], Arnaud Zaslavsky[3], Kristina Rackovic[3,4], Amalie Gjelsvik[1], David Pisa[5], Jan Soucek[5], and Ingrid Mann[1]

[1]Department of Physics and Technology, UiT The Arctic University of Norway, 9037, Tromsø, Norway
[2]Department of Surface and Plasma Science, Charles University Prague, 18000, Prague, Czech Republic
[3]LESIA – Observatoire de Paris, Université PSL, CNRS, Sorbonne Université, Université de Paris, 5 place Jules Janssen, 92195, Meudon, France
[4]Department of Astronomy, Faculty of Mathematics, University of Belgrade, Studentski trg 16, 11000, Belgrade, Serbia
[5]Institute of Atmospheric Physics, Czech Academy of Sciences, Bocni II/1401, 141 00 Prague, Czech Republic

**Correspondence:** Andreas Kvammen (andreas.kvammen@uit.no)

**Abstract.** This article present results from automatic detection of dust impact signals observed by the Solar Orbiter – Radio and Plasma Waves instrument.

A sharp and characteristic electric field signal is observed by the Radio and Plasma Waves instrument when a dust particle impact the spacecraft at high velocity. In this way, ∼5–20 dust impacts are daily detected as the Solar Orbiter travels through

the interstellar medium. The dust distribution in the inner solar system is largely uncharted and statistical studies of the detected dust impacts will enhance our understanding of the role of dust in the solar system.

It is however challenging to automatically detect and separate dust signals from the plural of other signal shapes for two main reasons. Firstly, since the spacecraft charging causes variable shapes of the impact signals and secondly because electromagnetic waves (such as solitary waves) may induce resembling electric field signals.

In this article, we propose a novel machine learning-based framework for detection of dust impacts. We consider two different supervised machine learning approaches: the support vector machine classifier and the convolutional neural network classifier. Furthermore, we compare the performance of the machine learning classifiers to the currently used on-board classification algorithm and analyze one and a half year of Radio and Plasma Waves instrument data.

Overall, we conclude that classification of dust impact signals is a suitable task for supervised machine learning techniques. In

particular, the convolutional neural network achieves a 96% ± 1% overall classification accuracy and 94% ± 2% dust detection precision, a significant improvement to the currently used on-board classifier with 85% overall classification accuracy and 75% dust detection precision. In addition, both the support vector machine and the convolutional neural network detects more dust particles (on average) than the on-board classification algorithm, with 14% ± 1% and 16% ± 7% detection enhancement respectively.

The proposed convolutional neural network classifier (or similar tools) should therefore be considered for post-processing of the electric field signals observed by the Solar Orbiter.



# 1 Introduction

## 1.1 The Dust Population in the Inner Solar System

The interplanetary dust population in the inner solar system ($\leq 1$ AU) is formed by collisional fragmentation of asteroids, comets and meteoroids. The meteoroids and the larger dust particles are in bound orbits around the Sun and their lifetime is limited by collisions, while the smaller particles that form through collisional fragmentation are repelled from the Sun by the radiation pressure force. The sources and sinks of the interplanetary dust particles are well-measured at the orbit of Earth, while there are few observations inside 1 AU.

Model calculations show that the number density of dust within 1 AU is diminished by collisional destruction (Ishimoto, 2000). However, there are a number of uncertainties that enter the model calculations since the dust collision rates depend both on the dust number density distribution and on the relative velocities between the dust particles. These parameters are generally unknown inside the orbit of the Earth and the estimated sizes of the fragmented dust particles are currently based on empirical relations, inferred from laboratory measurements of accelerated dust particles (Mann and Czechowski, 2005). Furthermore, there is an additional dust population with interstellar origin that stream through the solar system. The interstellar dust distribution is largely unknown and thus complicates the analysis of the stellar dust population. Remote observations of the zodiacal light and the Fraunhofer corona (F-corona) provide some information of the dust population dust within 1 AU, but mainly of the larger ($> \mu$m) dust particles (Mann et al., 2004). For all these reasons, in-situ measurements are needed in order to better understand the role of dust in the inner solar system.

## 1.2 Exploration of the Inner Solar System

At present, the inner solar system is explored by the Parker Solar Probe (Szalay et al., 2020), launched August 12, 2018, and the Solar Orbiter (Müller et al., 2020), launched February 10, 2020. While systematic studies of the dust flux near 1 AU are conducted with the Solar Terrestrial Relations Observatory (STEREO) (Zaslavsky et al., 2012) and Wind (Malaspina et al., 2014). The first analyses show that a large fraction of the observed dust particles are repelled from the Sun, i.e. the dust particles are in unbound orbits (Zaslavsky et al., 2021; Szalay et al., 2020; Malaspina et al., 2020). Mann and Czechowski (2021) used model calculations to explain the impact rates observed by the Parker Solar Probe with dust particles in unbound orbits with sizes larger than ∼100 nm. Mann and Czechowski (2021) modeled the dust production by collisional fragmentation near the Sun and the dust trajectories were calculated with included radiation pressure and Lorentz force terms. Mann and Czechowski (2021) showed that the observed impacts largely agrees with the model calculations for dust > 100 nm and the differences are possibly due to the influence of smaller particles, of local and temporal variations and of other dust components, such as dust in bound orbits and interstellar dust.

In this work, we analyze data acquired by the Solar Orbiter. The spacecraft orbits the Sun in an elliptic orbit with a period of approximately 6 months. At perihelion, the Solar Orbiter reaches a minimum solar distance of 0.28 AU, just within the





perihelion of the Mercury orbit. The expected mission duration is 7 years, with a possible 3 year extension. The Solar Orbiter
will thus provide long-term, in-situ observations of the environment in the inner solar system with multiple instruments. One
of these instruments is the Radio and Plasma Waves instrument, allowing observations of the cosmic dust flux with typical
diameters ranging from ∼100 nm to ∼500 nm (Zaslavsky et al., 2021).

### 1.3 Radio and Plasma Waves Instruments for Dust Detection

Radio and plasma waves instruments (i.e. antennas) have been used for studying dust in the solar system since the Voyager
mission (Gurnett et al., 1983; Aubier et al., 1983). A dust impact is observed by the spacecraft antennas as a sharp and charac-
teristic electric field signal, produced by the impact ionization process.

The impact ionization process occur when dust particles hit a target in space with impact speeds on the order of ∼km/s or
larger, impact speeds which are typical for space missions in the interplanetary medium. The kinetic energy of the impact is
transferred into deformation, shattering, melting and vaporization of the dust projectile– and target material, producing a cloud
of free electrons and ions on the spacecraft surface. Laboratory measurements (Collette et al., 2014) and model calculations
(Hornung et al., 2000) indicate that the free-charge yield depends on multiple parameters, where the most important are the
dust impact velocity, the dust mass and the material of both the dust projectile and the target (the spacecraft surface) (Mann
et al., 2019). The forming cloud of charged particles is partly expanding into the ambient solar wind and is partly recollected
by the spacecraft. This induces the characteristic electric field signal, hereafter called a dust impact signal/waveform.

Radio and plasma waves instruments allow for the the entire spacecraft body to serve as a dust detector, providing a large
collection area in comparison to dedicated dust instruments. Thus, radio and plasma waves instrument can provide dust distri-
bution estimates based on thousands of dust impacts each year, statistical products that are difficult to acquire by dedicated dust
instruments. Still, the radio and plasma waves instruments have lower sensitivities than dedicated dust detectors (Zaslavsky,
2015) and the shape of the dust impact waveform is highly dependent on the potential difference between the spacecraft and the
ambient plasma (Vaverka et al., 2017). This complicates the analysis of the dust distribution in the solar system since statistical
studies rely on automatic dust impact detection software with high accuracy.

### 80 1.4 Machine Learning Classification of Time Series Data

In this article, we present a machine learning-based framework as a novel method for detecting dust impact signals in radio
and plasma waves instrument data. Machine learning methods, in particular neural networks in the recent decade, have been
extensively used for challenging time series classification problems, such as: speech recognition (Trosten et al., 2019), heart
rate monitoring (Wickstrøm et al., 2022) and human activity classification (Villar et al., 2016).


A neural network has previously been used for selecting the signals of interest observed by the WAVES instrument on board
the Wind spacecraft (Bougeret et al., 1995). While an unsupervised method (self-organizing maps) was used for identifying and

categorizing plasma waves in the magnetic field data observed by the MMS$_1$ spacecraft (Vech and Malaspina, 2021). Still, no machine learning tools have been developed for classifying dust impacts in radio and plasma waves instrument data, although the characteristic signal produced by the impact ionization process is distinctive and could therefore be suitable for machine learning detection.

### 1.5 Motivation and Article Structure

The main purpose of this work was to develop a dedicated dust detection tool that can be used to automatically process the large amount of data acquired by the Radio and Plasma Waves instrument on board the Solar Orbiter. The aim was to develop a classifier with a high overall classification accuracy on a balanced data set that would make statistical studies more reliable and easier to conduct. For this project, we defined high accuracy to be ($\gtrsim 95\%$) after some initial testing. We considered ($\gtrsim 95\%$) accuracy to be satisfactory for statistical studies and a significant improvement to the currently used classification system. In order to achieve this objective we used supervised machine learning techniques to develop the dust classifiers, trained and tested on a set of 3000 manually labeled observations.

The remaining of this article is structured as follows. Section 2 explains the Solar Orbiter – Radio and Plasma Waves observations and the on-board algorithm that is currently used for dust impact detection. Section 3 describes the procedure that was used for developing the machine learning classifiers; from the downloaded data to the training– and testing of the classifiers. Section 4 investigate the performance of the classifiers and includes the resulting dust impact rates, calculated by analyzing one and a half year of automatically classified Solar Orbiter data. Finally, Section 5 presents the overall conclusions of this project.

## 2 Observations and Data Acquisition

### 2.1 The Radio and Plasma Waves (RPW) Instrument and the Time Domain Sampler (TDS) Receiver

This work focuses on electric field signals (i.e. waveforms) observed by the Radio and Plasma Waves (RPW) instrument on-board the Solar Orbiter (Maksimovic et al., 2020). The RPW instrument consist of 3 antennas operating synchronously and the measured electric potential is recorded by the Time Domain Sampler (TDS) receiver unit (Soucek et al., 2021).

The TDS receiver is designed to capture plasma waves (such as ion-acoustic and Langmuir waves) in the frequency range 200 Hz – 100 kHz, in addition to the dust impact signals (Soucek et al., 2021). The antenna voltages are converted to electric field values using the antenna effective lengths, but are otherwise uncalibrated. We consider only signals sampled with a sampling rate of 262.1 kHz in snapshots of 16384 time steps, acquired when the TDS receiver was operating in the XLD1 mode.



The XLD1 mode is the most commonly used observational mode of the RPW-TDS system (Soucek et al., 2021). XLD1 is a hybrid mode, where channel 3 ($CH_3$) is operating in monopole mode while channel 1 ($CH_1$) and channel 2 ($CH_2$) are operating in dipole mode:

$$CH_1 = \left(\frac{V_1 - V_{SC}}{L_1}\right)\hat{L}_1 - \left(\frac{V_3 - V_{SC}}{L_3}\right)\hat{L}_3 \tag{1}$$

$$CH_2 = \left(\frac{V_2 - V_{SC}}{L_2}\right)\hat{L}_2 - \left(\frac{V_1 - V_{SC}}{L_1}\right)\hat{L}_1 \tag{2}$$

$$CH_3 = \left(\frac{V_2 - V_{SC}}{L_2}\right)\hat{L}_2 \tag{3}$$

where $V_i - V_{SC}$ denotes the potential difference between antenna $i$ and the spacecraft body along the antenna boom with unit vector $\hat{L}_i$ and effective length $L_i$. For this work however, the 3 RPW antenna signals are all converted to monopole electric field signals ($\bar{E}_1, \bar{E}_2, \bar{E}_3$) by the following conversion:

$$\bar{E}_1 = CH_3 - CH_2 = \left(\frac{V_2 - V_{SC}}{L_2}\right)\hat{L}_2 - \left(\left(\frac{V_2 - V_{SC}}{L_2}\right)\hat{L}_2 - \left(\frac{V_1 - V_{SC}}{L_1}\right)\hat{L}_1\right) = \left(\frac{V_1 - V_{SC}}{L_1}\right)\hat{L}_1 \tag{4}$$

$$\bar{E}_2 = CH_3 = \left(\frac{V_2 - V_{SC}}{L_2}\right)\hat{L}_2 \tag{5}$$

$$\bar{E}_3 = CH_3 - CH_2 - CH_1 = \bar{E}_1 - CH_1$$
$$= \left(\frac{V_1 - V_{SC}}{L_1}\right)\hat{L}_1 - \left(\left(\frac{V_1 - V_{SC}}{L_1}\right)\hat{L}_1 - \left(\frac{V_3 - V_{SC}}{L_3}\right)\hat{L}_3\right) = \left(\frac{V_3 - V_{SC}}{L_3}\right)\hat{L}_3 \tag{6}$$

## 2.2 The Triggered Snapshot WaveForms (TSWF) data product and the TDS Classifier

For this project, we use the Triggered Snapshot WaveForms (TSWF) data product, processed with software version 2.1.1 and acquired over a one and a half year period, spanning between June 15, 2020, to December 16, 2021. The TSWF data product consists of signal packets (63 ms snapshots) that are downlinked only if the classification algorithm on-board the Solar Orbiter is triggered. The accuracy of the on-board classification algorithm is therefore important in order to optimize the data transfer and provide reliable data products for statistical analysis.

The input to the on-board classification algorithm, hereafter named the TDS classifier or the TDS classification algorithm, is the 63 ms signal packet, while the output is categorized into one out of three labels: *dust*, *wave* or *other*. The TDS classifier assigns the label based on 3 extracted features.

1. The snapshot peak amplitude

2. The ratio of the peak amplitude to the median of the signal

3. The bandwidth of the main spectral peak identified in the Fourier spectrum

The signal label is then determined by comparing the extracted feature values against configurable thresholds. For more detailed descriptions of the TDS classifier, see Maksimovic et al. (2020) and Soucek et al. (2021). Figure 1 presents a few examples of





recorded snapshots with included labels, as classified by the TDS classification algorithm.

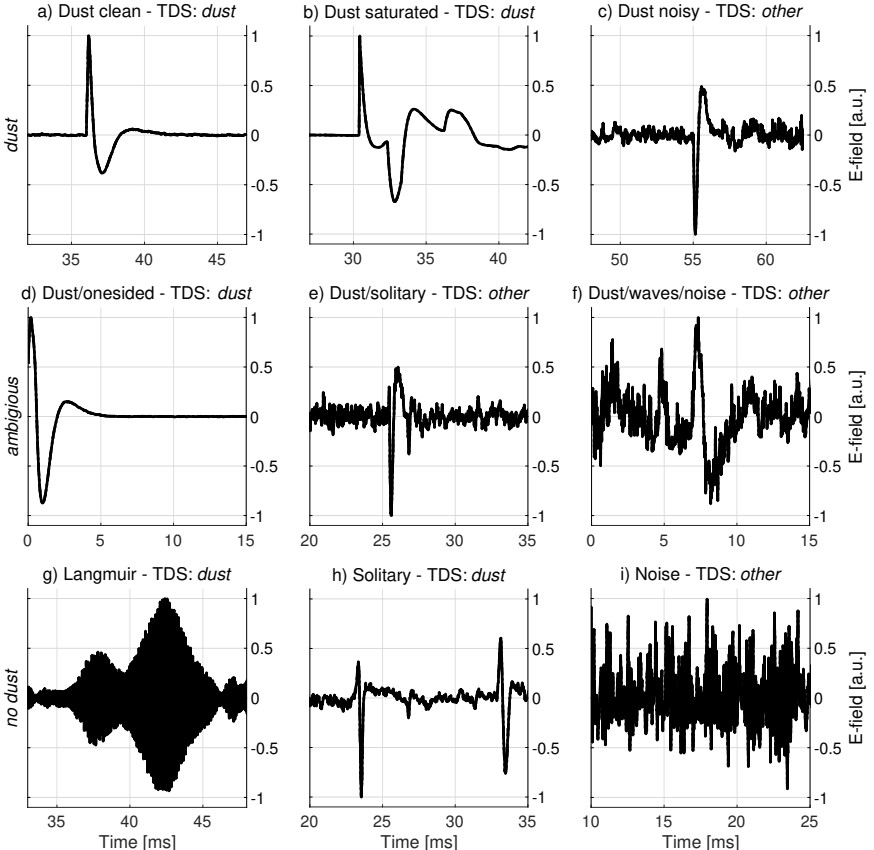

**Figure 1.** Waveforms recorded by the TDS receiver and measured by one of the RPW antennas. The signal label, classified by the TDS classification algorithm, is included for each snapshot in the subplot titles. The top row presents dust waveforms: a) is a clean dust impact waveform, b) shows a dust impact that saturates the receiver unit (or reaches the non-linearity limit), c) presents a weak dust impact signal that is strongly affected by noise. The middle row presents ambiguous waveforms: d) might be a dust impact, but information is limited by the signal framing, e) is likely a dust impact, but the signal shape resembles solitary waves and is strongly affected by noise, f) might be a dust impact, but noise and possible electromagnetic waves makes the signal difficult to interpret. The bottom row presents waveforms without dust: g) shows Langmuir waves, characterized by the high-frequency E-field oscillations with a lower-frequency amplitude modulation, h) presents solitary waves, which sometimes resemble dust impact waveforms, i) shows a signal dominated by noise, without any clear features. Note that the full (63 ms) snapshots are zoomed to 15 ms intervals around the interesting features and that the signal amplitudes are normalized to ±1 and centered around zero for illustrative purposes.





Figure 1 illustrates that it is challenging to detect and separate dust signals from the plural of other signal shapes. In particular, the dust waveform in Sub-figure c) is classified as *other*, while the Langmuir wave and solitary wave snapshots in Sub-figures g) and h) are erroneously classified as *dust* by the TDS classification algorithm.

## 3 Machine Learning-Based Framework for Automatic Dust Impact Detection

The goal of the machine learning classifier is to take a monopole RPW snapshot as an input and automatically output if the signal contains a dust impact or not. For this purpose, we use a supervised classifier. A supervised classifier relies on manually labeled data to learn (i.e. train) the function that maps the input observation (the electric field signal) to the output label. For this work, we focus on detecting dust impact signals, we therefore use a binary label: *dust* or *no dust*. Additional labels, such as: *ion-acoustic waves*, *Langmuir waves* and *solitary waves*, could however be implemented in a similar machine learning-based framework.

### 3.1 Data Pre-Processing for Machine Learning Classification

In order to construct a balanced data set, we selected $\sim$ 1500 waveforms classified as *dust* and $\sim$ 1500 waveforms classified as *wave/other* by the TDS classification algorithm. The signals were randomly drawn from the TDS data archive and acquired between 15 June 2020 to 16 December 2021. The TDS signals were then pre-processed in order to standardize the input to the classifier and speed up the training. Standardized data further reduces bias effects and makes the manual labeling of the signals easier to conduct. For this work, a 4-step pre-processing procedure was used independently on each antenna signal, the pre-processing procedure applied on a sample signal is illustrated in Figure 2.

1. **Remove the signal offset** The electric field offset is removed by subtracting the raw signal with the median of a heavily filtered version of the raw data. A sliding median filter over 21 time steps was selected by visual inspection of the noise characteristics. The removal of the electric field offset centers the signal around zero and reduces bias effects from offset waveforms.

2. **Filter the data** The signal is filtered using a sliding median filter over 7 time steps in order to reduce the high-frequency noise. The 7 time step filter was selected by inspecting the power spectrum of impact signals and by noticing that most information above ($f_N$ = 35 kHz) is buried in noise, although the TDS sampling frequency is higher ($f_s$ = 262.1 kHz), thus making a filter length ($< f_s/f_N \approx 7.5$) appropriate without significant loss of information.

3. **Compress the data** The signal is re-sampled with a compression factor of 4 using linear 1-dimensional interpolation. The compression is done to speed up the training of the classifier, resulting in a re-sampling from 16384 to 4096 time steps.

4. **Normalize the signal** The data is normalized to be between -1 and 1 by dividing all data samples with the maximum absolute value of the signal. The normalization makes the machine learning classifier more robust to variations in the signal strength and eases the parameter optimization.

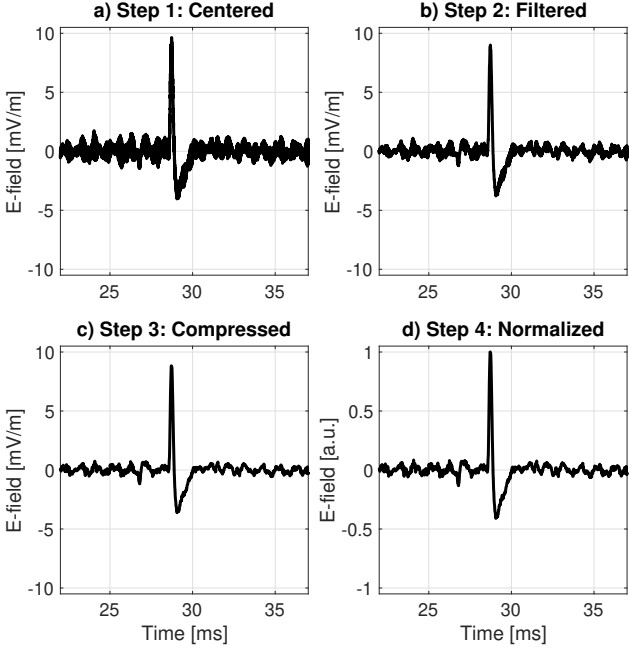

**Figure 2.** A dust waveform observed by antenna 2 on September 8, 2021. The sub-figures illustrate the different stages of the pre-processing procedure. a) The electric field offset is removed and the signal is centered around 0 mV/m. b) The signal is filtered by a median filter over 7 time steps to reduce the high-frequency noise. c) The signal is compressed by a factor of 4 to reduce the data size. d) The waveform is normalized by the maximum absolute value of the signal in order to ease the parameter optimization of the machine learning classifier. Note the waveform is zoomed to a 15 ms time period around the dust impact in order to better visualize the impact shape modification by the pre-processing procedure.

## 3.2 Manual Waveform Labeling

Manually labeled data is used both to train the machine learning classifiers and to test the performance of the trained models. Thus, great care is needed in order to construct a high-quality labeled data set, without significant contamination of corrupted data files, biases and mislabeled signals.

We manually labeled the data into either *dust* or *no dust*. Each signal was displayed without indications of the previously
assigned label by the TDS classifier in order to reduce bias effects. Furthermore, a zoom function was used to investigate the areas of interest and options were included both to correct labeling mistakes by the user and to indicate ambiguous signals that do not clearly fit into any label (*dust* or *no dust*). Appendix A presents the Graphical User Interface (GUI) that was used to label the 3000 observations.





It should be noted that 134 signals (i.e. 4.5%), out of 3000 manually labeled waveforms, were marked as ambiguous and did not clearly fit into either the *dust* or *no dust* label, see Figure 1 for ambiguous examples. Furthermore, the manual waveform labeling was done by one scientist, although with consultations with other experts. Thus, it is to be expected that different scientists will disagree on a proportion (around 5%) of the the manual labels. The disagreement level could possibly be reduced if several experts labeled the same data set and the labeling consensus was used as the effective waveform label.

## 3.3    Developing the Machine Learning Classifiers

The manually labeled data was split into a training set (containing 80% of the data) and a testing set (with the remaining 20%). The training data is used to optimize the free parameters of the machine learning classifier with respect to the assigned labels, while the testing data is used as an independent set to test the performance of the trained classifiers. The performance of a machine learning classifier is quantified by comparing the outputs of the trained model to the labels of the testing data. Figure

3 illustrates the data flow; from the TDS data sets to the machine learning performance metrics.

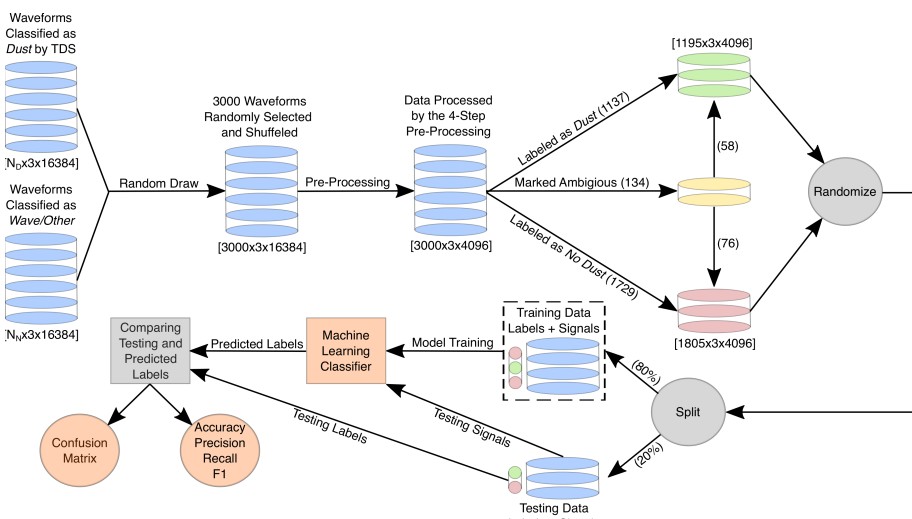

**Figure 3.** Data flow: from the TDS data sets to the machine learning performance metrics. The diagram illustrates the data flow by the black arrows and the applied process by the arrow label. The cylinders indicate the signal waveforms and the cylinder color indicate the associated label. The gray circles mark data transformation processes. The random draw of the TDS data and the pre-processing is explained in Sub-section 3.1, while the manual labeling is described in Sub-section 3.2. The randomization and splitting of the manually labeled data into a training and a testing set is described in Sub-section 3.3 and the training and testing of the machine learning classifiers is explained in Sub-sections 3.4 and 3.5. Finally, the performance of the machine learning classifiers are compared and evaluated in Sub-section 4.1.

There are numerous machine learning techniques that are suitable for time series classification. In this work, we focus on two well-known techniques: the Support Vector Machine (SVM) and the Convolutional Neural Network (CNN).





## 3.4 The Support Vector Machine (SVM)

The support vector machine (Boser et al., 1992; Cortes and Vapnik, 1995) is a robust and versatile classification algorithm, considered to be one of the most influential approaches in supervised learning (Goodfellow et al., 2016). SVMs learn the decision hyperplane that maximizes the discriminative power between the observations categorized into two classes (in this case: *dust* or *no dust*). However, SVMs are highly dependent on the representation of the data and often achieve sub-optimal performance on high-dimensional data (when used directly). In this case, the observation from 3 antenna measurements, each

with 4096 time steps, is both high dimensional and noisy (each time step contain little information). It is therefore common to extract important characteristics (i.e. features) from the data to provide the SVM with compactly represented information with less noise and redundancies.

### 3.4.1 Feature Extraction

In order to develop a baseline machine learning classifier, comparable to the on-board TDS classification algorithm, a 2-dimensional SVM classifier was considered. Thus, every observation with dimension (3x4096) is represented by a 2-dimensional feature vector (1x2). After some initial testing, we selected two features that had a high discriminative power between the *dust* and *no dust* observations.

1. **The standard deviation** The mean standard deviation is calculated over the 3 antenna channels, each with 4096 time steps. The standard deviation is appropriate since normalized *dust* signals typically have a lower mean standard deviation than normalized *no dust* signals.

    2. **The convolution ratio** The $\log_{10}$ value of the convolution ratio ($|\text{conv}|_{\max}/|\text{conv}|_{\text{median}}$) is calculated, where |conv| is the absolute values of the convolution of the antenna signals with a normalized Gaussian of width 0.5 ms. $|\text{conv}|_{\max}$ is

the maximum value of $|\text{conv}|$, while $|\text{conv}|_{\text{median}}$ is the median. The convolution ratio was selected as a feature since the *dust* signals typically have a larger convolution ratio than the *no dust* signals. The Gaussian width of 0.5 ms was experimentally found to give high correlations with dust impact signals.

### 3.4.2 Training the Support Vector Machine

The 2 features (standard deviation and convolution ratio) were extracted from all observations in the training data. The decision

hyperplane, in this 2-dimensional case a decision line, is defined by a polynomial of degree 2 that is optimized by minimizing the non-separable SVM cost function, see e.g. Theodoridis and Koutroumbas (2009) for details. The SVM classifier was trained with a slack variable factor of 1 and equal weighting between the *dust* and *no dust* observations. Figure 4 illustrates the training of the SVM classifier.





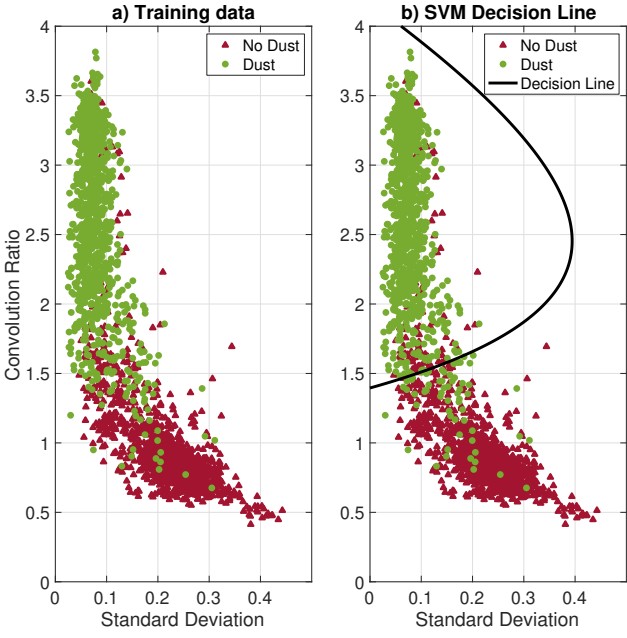

**Figure 4.** a) The (1x2) feature vectors extracted from all (2400) observations in the training data, the associated labels are indicated in green (*dust*) and red (*no dust*). b) The SVM decision line, the optimal second order polynomial, obtained by minimizing the non-separable SVM cost function. The SVM decision line appears to be reasonable and most observations are separable.

### 3.4.3 Testing the Support Vector Machine

The performance of the trained SVM classifier is evaluated using the independent testing data, i.e. the remaining manually labeled data (20 %) that was not used for training the classifier. Figure 5 presents the SVM classification performance on the testing data.

Overall, the SVM classifier achieves a classification accuracy of 94% on the testing data using the 2-dimensional feature vectors. Note that the inclusion of more extracted features could possibly enhance the SVM performance. Several additional features could be considered, such as; the mean amplitude of the signal, the range between the signal maximum and minimum values and the cross-correlation length (the time lag to the first zero crossing).

### 3.4.4 Explainability of the Support Vector Machine

Ideally, we want to develop a machine learning classifier that not only has a high accuracy, but also make decisions that are understandable for a human expert (Holzinger et al., 2019). In other words, we want to be able to explain why the machine learning classifier selected the predicted class for a given observation. In machine learning, this is often referred to as the explainability of the trained classifier. Figure 5 presents the testing data in the 2–D feature vector space, but this plot gives no





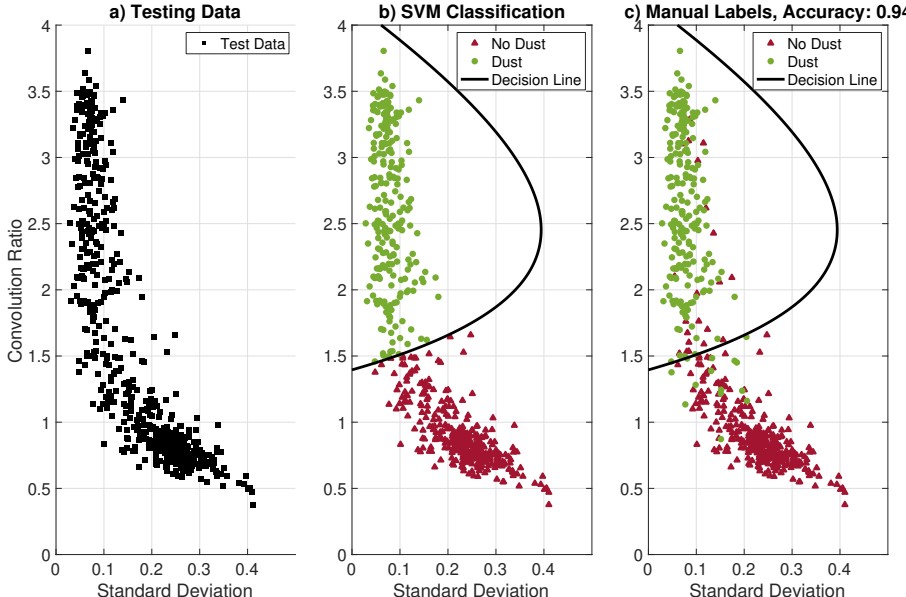

**Figure 5.** a) The (1x2) feature vectors extracted from the testing data (600 observations with hidden labels). b) The testing data is classified using the trained SVM decision line, where all observation within the polynomial line is classified as *dust* while all observations outside are classified as *no dust*. c) The "true" labels (from the manual labeling) are revealed. It is clear that some observations are confused, predominantly near the decision line. Still, the SVM classifier achieves an overall classification accuracy of 94%, calculated by comparing the outputs from the SVM classification (Sub-figure b) to the "true" labels (Sub-figure c).

clear indications of how different signal shapes are distributed and which signatures are confused by the SVM classifier. In
order to better understand the decisions made by the SVM classifier, the signal examples in Figure 1 are studied in detail. The analysis is presented in Figure 6.

It should be noted that the signal examples in Figure 6 are not representative for the general distribution of observations in the 2–D feature vector space, since most observations are clustered in distinct *dust* and *no dust* regions, as can be seen
in Figure 5. Figure 6 focuses mostly signal examples that are challenging to classify. Still, Figure 6 indicates that the SVM classifier provides mostly comprehensible outputs, but might have difficulties classifying signals with important signatures located at the edge of the snapshot frame.





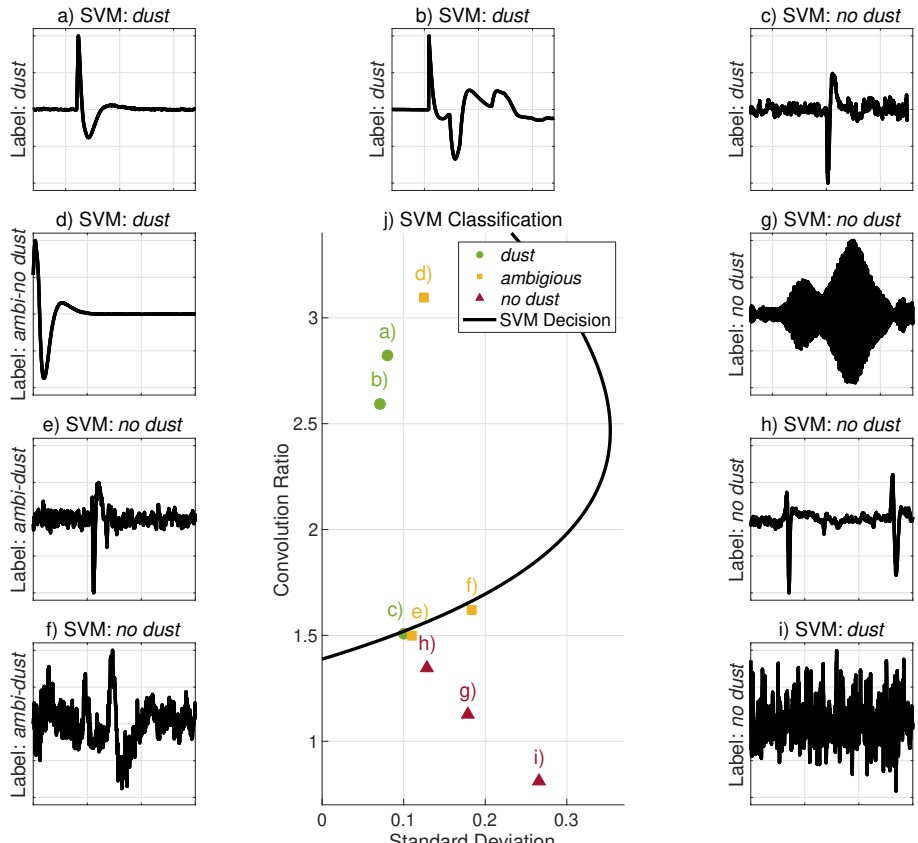

**Figure 6.** The signal examples are presented in Sub-figures (a-i), the manual labels are indicated along the y-axis and the predicted label, classified by the SVM decision line, are presented in the sub-plot titles. Sub-Figure j) presents the associated signal examples in the 2–D feature vector space along with the SVM decision line. The *dust* signals are illustrated in green, the *ambiguous* signals are marked in yellow and the *no dust* signals are indicated in red. The SVM classifier provides mostly explainable outputs. The clear dust signals (a-b) are located well within the SVM decision line, the *ambiguous* signals (e-f) are located near the decision line while the no dust signals (g-i) are clearly located outside. However, *dust* signal c) is erroneously located just outside the decision line, this can possibly be explained a weak signal-to-noise ratio. In addition, signal d) is located well within the decision line, although this signal is labeled *ambiguous-no dust* due to the signal framing, this indicates that the SVM might have difficulties classifying signatures located at the edge of the snapshot frame.





## 3.5 The Convolutional Neural Network (CNN)

Convolutional Neural Networks are algorithms designed for processing grid-like data and have achieved premium performance

on a number of different tasks in the recent decade, such as image (He et al., 2016; Kvammen et al., 2020), video (Karpathy et al., 2014), and time series (Wang et al., 2017; Wickstrøm et al., 2021) classification.

### 3.5.1 Feature Extraction

Unlike the SVM, the CNN do not require pre-defined feature extraction routines. Instead, the CNN extracts the features based on a chain of convolution operations and automatically optimizes the convolution filters based on the training data and the

associated labels.

For this work, we employed the 3-layer fully convolutional network architecture presented in Wang et al. (2017) and suggested for time series classification after extensive testing (Wickstrøm et al., 2022; Fawaz et al., 2020; Karim et al., 2019). The Rectified Linear Unit (ReLU) function (Glorot et al., 2011) was used as the activation function and Batch Normalization (BN)

(Ioffe and Szegedy, 2015) was used at each convolutional layer in order to regularize the network and accelerate the training process. Figure 7 presents the employed CNN architecture.

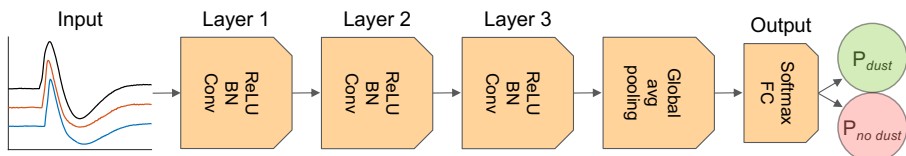

**Figure 7.** The 3-layer fully convolutional network used for dust impact classification. The input to the network is the (3x4096) waveform. The function that maps the input waveform to the output label: *dust* or *no dust* is defined by 3 convolutional layers, consisting of 128, 256 and 128 independent filters with kernel lengths of 8, 5 and 3 weights, respectively. Batch normalization (BN) is used at each convolutional layer to regularize the the inputs and the Rectified Linear Unit (ReLU) function was used as the activation function. Finally, the output of the convolutional layers (with dimension 128x4096) is averaged in the global pooling layer to a vector with dimension (128x1). The class score is then determined in a Fully Connected (FC) network layer and the output label probabilities ($P_{dust}$, $P_{no\ dust}$) are calculated using the softmax function. The Figure is adopted from Wickstrøm et al. (2021).

### 3.5.2 Training the Convolutional Neural Network

The 3-layer fully convolutional network consists of 267010 free parameters (weights and biases) that need to be optimized to solve the dust impact classification task. The free parameters are randomly initialized and thereafter optimized using the

ADAM gradient descent optimizer (Kingma and Ba, 2014). The CNN was trained for 225 epochs with a cross-entropy loss function using the 2400 labeled observations in the training data. For more details on neural network training and optimization, see for example (Montavon et al., 2012).



### 3.5.3 Testing the Convolutional Neural Network

In order to visualize the features extracted by the CNN, we employ the t-distributed Stochastic Neighbor Embedding (t-SNE)
method (Van der Maaten and Hinton, 2008). The t-SNE method is used for visualizing high-dimension data by assigning each
observation a location in a 2–D space such that similar observations are modeled by nearby points while dissimilar observations
are modeled by distant points with high probability. The (128x1) testing feature vectors, extracted in the global pooling layer,
are presented in a 2–D t-SNE map in Figure 8, along with a visualization of the CNN classification performance.

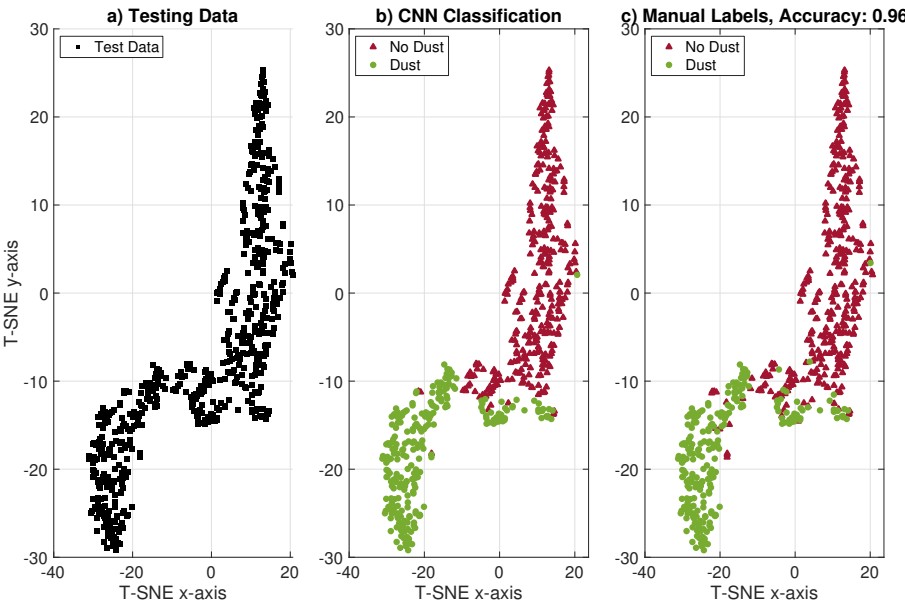

**Figure 8.** a) The testing data visualized by a dimension-reduced t-SNE map where similar feature vectors are modeled by nearby points while
dissimilar observations are modeled by distant points with high probability. b) The testing data classified by the trained CNN. c) The "true"
manual labels are presented. Only a few observations, predominantly in the transition region between the *dust* and *no dust* observations are
confused. An overall classification accuracy of 96% is calculated by comparing the labels predicted by the CNN to the manual labels.

Overall, the CNN obtains a high ($\gtrsim 95\%$) classification accuracy and might therefore be suitable for automatic processing
of electric field signals observed by the RPW instrument on board the Solar Orbiter.

### 3.5.4 Explainability of the Convolutional Neural Network

Neural networks have traditionally been regarded as black boxes (Shwartz-Ziv and Tishby, 2017; Alain and Bengio, 2016),
where the network carries out the desired task, but the network decisions are difficult to interpret. However, progress have
been made in recent years for making the neural network decisions more accessible and easier to interpret (i.e. explainable) for
human users (Samek et al., 2021). In this section, we analyze the CNN decisions by employing Class Activation Maps and the



previously described t-SNE method.

Class Activation Maps (CAMs) (Zhou et al., 2016) highlights the regions of the data that are important for a considered
label ($c$) by analyzing the features extracted in the global pooling layer and the weights in the FC layer that are associated with
label ($c$), see e.g. (Wang et al., 2017) for a detailed description. The outcome of the CAM analysis is that we can visualize
the sections of the signal that are influential for the CNN classification decision. Figure 9 presents the CAM analysis of the
signal examples in Figure 1 along with an illustration of the signal features in a dimension-reduced t-SNE space. Note that the
t-SNE mapping in Figure 9 is different from the t-SNE mapping in Figure 8, since Figure 9 considers a different CNN where
the signal examples are excluded from the training data.

The CAM values in Figure 9 illustrate that the CNN make classification decisions that are comprehensible (in most cases).
It is however interesting to note that signal c), manually labeled as *dust*, is erroneously classified as *no dust* by the CNN, and
that this decision is largely based on the tail (the relaxation period) of the impact signal. It should however be noted that it is
more difficult to explain the *no dust* predictions than the *dust* predictions since the *no dust* CNN decisions are based on the lack
of a signature (*dust* impact), rather than on the presence of signature. In addition, signal d), manually labeled as *ambiguous-
no dust*, is classified as *dust* by the CNN, and this decision is based on a wide region of the signal with emphasis on the tail
of the (ambiguous) dust impact signal, this section might not have been highlighted as particularly important by a human expert.

In general however, the CNN achieves a high accuracy (>95%) and make decisions that are mostly in-line with human
interpretation. It is therefore reasonable to infer that the CNN will have a performance comparable to the agreement level
between human experts, where disagreement predominantly occurs for ambiguous and noisy signals, while clear *dust* and clear
*no dust* signals are classified correctly.



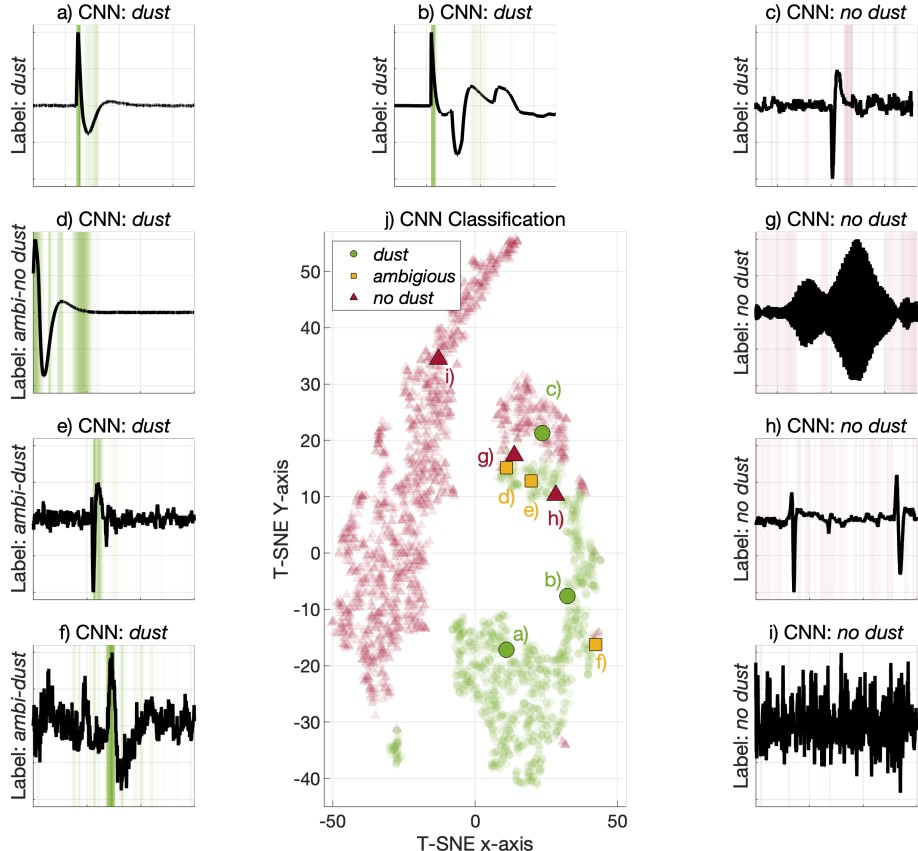

**Figure 9.** The signal examples and the CAM analysis are presented in Sub-figures (a-i), the manual labels are indicated along the y-axis and the predicted label, classified by the CNN, is presented in the sub-plot titles. Sub-figure j) presents the associated signal examples in the t-SNE space along with the training data signals as transparent points. The *dust* signals are illustrated by the green dots, the *ambiguous* signal examples are marked in yellow and the *no dust* signals are indicated in red. The CAM analysis show that the CNN emphasise the *dust* impact sections similarly to human experts, where the highlighted green regions indicate positive CAM values. Also the *no dust* CAM values (highlighted in red) are mostly understandable, although it is difficult to interpret the CNN decisions that are based on the lack of a signature (*dust* impact), rather than on the presence of signature. The t-SNE map show that the clear *dust* signals (a-b) are distinctly located in a green (*dust*) region whereas the clear *no dust* signal i) is distinctly located in a red (*no dust*) region. The remaining signals are located in more mixed regions. It should however be noted that the observations are represented by a 128 dimensional feature vector in the CNN and that the (2–D) t-SNE representation diminishes a lot of information, meaning that even the signals located in a mixed region of the t-SNE plot might be separable in the 128 dimensional feature vector space.



## 4 Results and Discussions

### 4.1 The Average Classification Performance Metrics


The average classification performance is obtained by training and testing the machine learning classifiers on 10 runs, each run with different training and testing sets. The classifiers are initialized from scratch and the testing and training sets are selected independently 10 times by randomization and splitting of the manually labeled data, as indicated by the gray circles in Figure 3. The average class-wise performance of the on-board TDS classifier and the machine learning SVM and CNN classifiers are summarized as confusion matrices in Figure 10. Overall, the CNN has the highest performance for both *dust* and *no dust* classification. In addition, both the SVM and the CNN obtain stable performance with only small variations for each run.

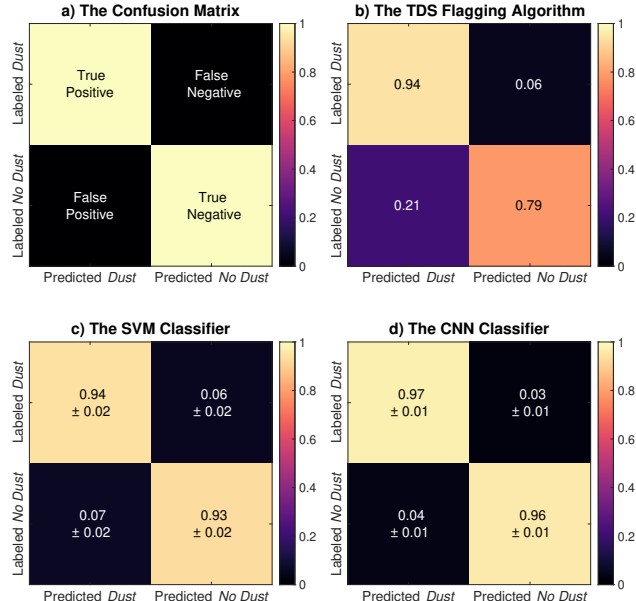

**Figure 10.** a) The confusion matrix entries are described by the true (correctly classified) and false (erroneously classified) values, as compared to the manual labels, positive indicate *dust* predictions and negative indicate *no dust* predictions. b) The TDS classifier confuses *dust* and *no dust* observations, where a large proportion ($> 0.20$) of *dust* predictions are manually labeled as *no dust*. c) The SVM classifier predicts both *dust* and *no dust* observations with a high ($> 0.90$) accuracy. d) The CNN classifier predicts a very large ($> 0.95$) proportion of both *dust* and *no dust* observations correctly.

The classification performance is further evaluated by the accuracy, precision, recall and F1 score. The definitions for the performance metrics are included in Appendix B. The average performance metrics, calculated over 10 runs, are summarized in Table 1. Again, the CNN has the highest performance across all metrics. Furthermore, the CNN obtain a significant improvement in the classification performance with a statistical significance at a level of 0.01, computed using a t-test. The t-test was





computed in a pairwise manner between both the CNN and the SVM, and the CNN and the TDS. In all cases, the enhanced performance of the CNN classifier was significant.

**Table 1.** The TDS, SVM and CNN classification performance metrics: accuracy, precision, recall and F1-score. The SVM and CNN scores and error values are the mean and the standard deviation across 10 training runs. The bold numbers indicate statistically enhanced performance with a significance level of 0.01, computed using a t-test.

| Classifier | Accuracy | Precision | Recall | F1 Score |
|---|---|---|---|---|
| TDS | 0.850 | 0.746 | 0.944 | 0.833 |
| SVM | $0.936 \pm 0.012$ | $0.903 \pm 0.027$ | $0.941 \pm 0.017$ | $0.921 \pm 0.015$ |
| CNN | $\mathbf{0.964 \pm 0.006}$ | $\mathbf{0.939 \pm 0.020}$ | $\mathbf{0.972 \pm 0.008}$ | $\mathbf{0.955 \pm 0.008}$ |


The results from both the confusion matrices and the performance metrics strongly suggest that the SVM and CNN classifiers provide binary classification results with a higher reliability than the TDS classifier. We therefore propose that the CNN classifier (or similar tools) should be considered for post-processing of the TDS data product in statistical studies of dust impacts observed by the Solar Orbiter RPW instrument. Finally, it should be noted that 134 signals (i.e. 4.5%), out of 3000

manually labeled waveforms, were marked as ambiguous, illustrated by the yellow cylinder Figure 3, and did not clearly fit into either the *dust* or *no dust* label, see Figure 1 for label examples. It is therefore improbable to achieve a classification accuracy exceeding ∼98%, and an accuracy approaching ∼99% should be considered suspicious and can be an indication of over-fitting.

### 4.2 The Dust Impact Rate

The trained classifiers can be used to automatically process large data sets. Figure 11 presents the TDS, SVM and CNN daily

impact rates, calculated by classifying all (∼82 000) monopole triggered waveforms acquired over a one and a half year period, spanning between June 15, 2020, to December 16, 2021. The impact rate function curve is obtained by fitting the dust flux model from Zaslavsky et al. (2021) (Equation 10) with an included offset:

$$R = F_{1\mathrm{AU}} S_{col} \left( \frac{r}{1\mathrm{AU}} \right)^{-2} \frac{\nu_{\mathrm{impact}}}{\nu_\beta} \left( \frac{\nu_{\mathrm{impact}}}{\nu_{\mathrm{impact}}(1\mathrm{AU})} \right)^{\alpha\delta} + C \qquad (7)$$

Where $F_{1\mathrm{AU}}$ is the unknown cumulative flux of particles above the detection threshold at 1 AU and $S_{col} = 8\mathrm{m}^2$ is the Solar

Orbiter collection area, as defined in Zaslavsky et al. (2021). Furthermore, $r$ is the radial distance from the sun, $\nu_{\mathrm{impact}}$ is the relative velocity between the spacecraft and the dust particles, assuming a constant radial and azimuthal velocity vector: $\nu_\beta$ = [50 km/s, 0 km/s], and the product $\alpha\delta$ = 1.3, as suggested in Zaslavsky et al. (2021). The assumed constant radial velocity is a good approximation for dust in hyperbolic orbits originating near the Sun that are deflected outward by the radiation pressure force. Finally, we included a constant impact rate offset: $C$, in order to obtain an improved fit. The description of the dust flux

in Equation 7 is based on the assumption that the dust– and spacecraft orbits are in the same orbital plane.




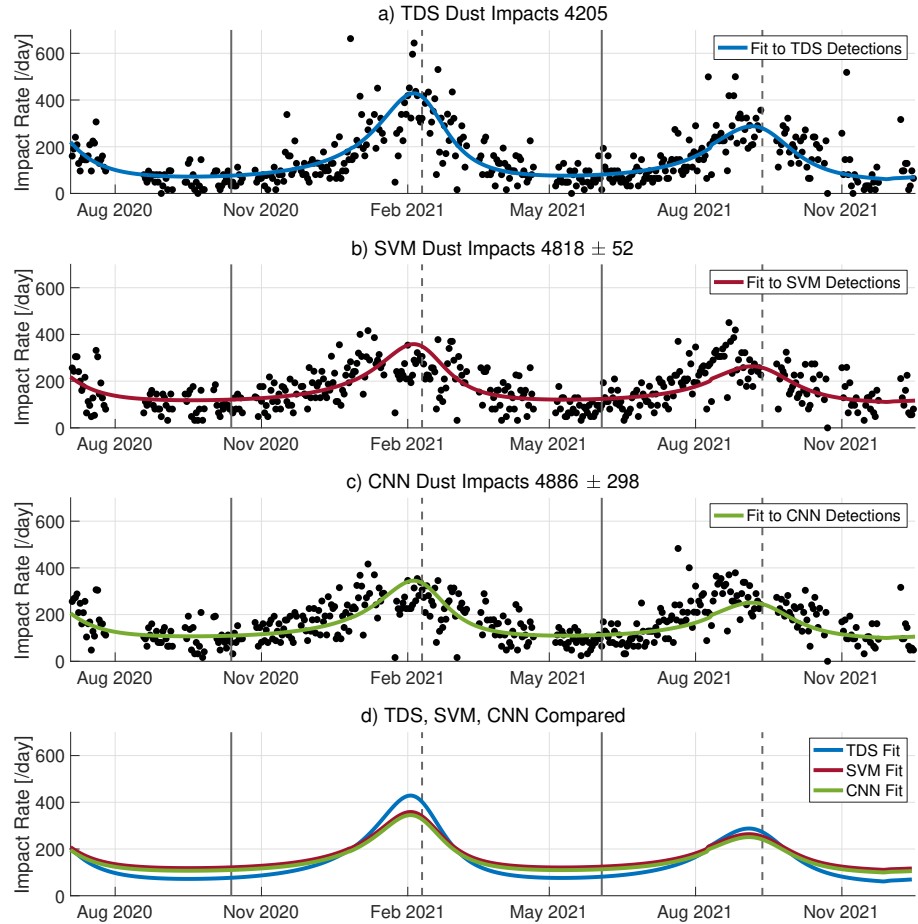

**Figure 11.** a) The daily dust impact rates according to the TDS classifier. The full vertical lines indicate times where the Solar Orbiter is at aphelion while the dashed lines indicate times at perihelion. b) The median of the daily impact rates classified by 10 trained SVM classifiers. c) The median of the daily impact rates from the 10 CNN classifiers. The impact rate function curves are obtained by fitting the dust flux model from Zaslavsky et al. (2021), Equation 7. d) The impact rate function cures are compared. The SVM and CNN dust impact rates are very similar, whereas the TDS provide notably smaller impact rates at aphelion and higher impact rates at perihelion. The daily impact rates are calculated from the daily dust impact number and the time dependent RPW duty cycle by assuming a constant impact probability for each day. The accumulated dust impact count for the TDS classification algorithm and the the mean and standard deviation of the accumulated dust impact count for the 10 CNN and SVM classifiers are presented in the sub-plot titles.



Figure 11 shows that the machine learning classifiers detected significantly more dust particles than the TDS classifier. The SVMs obtained a dust impact detection enhancement of 14% ± 1% while the CNNs had a 16% ± 7% increase. Both the SVM and the CNN classifiers obtain impact rates that are notably higher around the aphelion and distinctly lower in the vicinity of the perihelion, as compared to the dynamic range in the TDS dust impact rates.

Furthermore, Figure 11 illustrates that the fitted SVM and CNN impact rate function cures are in very good agreement. It is promising that two entirely different machine learning approaches provide comparable impact rates after classifying a large data set (consisting of ∼82 000 observations) when trained– and tested on a limited data set consisting of 3000 observations. This suggest that both the SVM and CNN classifiers have obtained stable performances and can be used to classify observations outside the domain of the training and testing data.

## 5 Conclusions

### 5.1 Summary and Scientific Implications

We have presented a machine learning-based framework for fully automated detection of dust impacts observed by the Solar Orbiter – Radio and Plasma Waves (RPW) instrument. Two different supervised machine learning approaches were considered: the Support Vector Machine (SVM) and the Convolutional Neural Network (CNN). The CNN classifier obtained the highest performance across all evaluation metrics and achieved 96% ± 1% overall classification accuracy and 94% ± 2% dust detection precision, a significant improvement to the currently used on-board TDS classification algorithm with 85% overall classification accuracy and 75% dust detection precision. We therefore conclude that the CNN classifier (or similar tools) should be considered for post-processing of the TDS data product for statistical studies of dust impacts observed by the Solar Orbiter.

The labeled data and the trained SVM and CNN classifiers are available online with included user instructions. The proposed method and the presented classifiers can thus provide the stellar dust community with thoroughly tested and more reliable data products than currently in use. It should also be noted that machine learning-based frameworks, similar to the SVM and CNN classifiers proposed in this article, can be developed for automatic processing of data acquired by radio and plasma waves instruments on-board other spacecrafts, such as: the Solar Terrestrial Relations Observatory (STEREO) (Zaslavsky et al., 2012), WIND (Malaspina et al., 2014), and the Parker Solar Probe (Szalay et al., 2020).

The SVM and CNN classifiers were used to process (∼82 000) uncalibrated monopole electric field signals acquired over a one and a half year period, spanning between June 15, 2020, to December 16, 2021. On average, the machine learning classifiers detected more dust particles than the currently used TDS algorithm, the SVMs had a 14% ± 1% detection enhancement and the CNNs had a 16% ± 7% increase. Furthermore, the SVM and CNN classifiers were in very good agreement and both classifiers obtained a notably higher dust impact rate in the vicinity of aphelion and a distinctly lower impact rate at perihelion,





as compared to the dynamic range of the TDS impact rates. This indicates a higher ambient dust distribution and/or a higher
radial dust velocity than previously observed. This result is significant since it implies the presence of other dust populations
in the data. Possible other populations are interstellar dust and interplanetary dust in bound orbits.

## 5.2    Outlook

The presented machine learning classifiers may be considered for on-board processing of the observed electric field signals.
However, the trained SVM and CNN classifiers presented in this article are trained on Triggered Snapshot WaveForms (TSWF)
data, and should not be used for processing 'untriggered" signals without additional training and testing on 'untriggered" data.
It should also be noted that the classifiers presented in this work are trained and tested on data labeled by one scientist,
although with consultations with other experts. Labeled data from several experts could provide machine learning classifiers
that are more in-line with the labeling consensus in the stellar dust community. Additional labeling can also be use to extended
the machine leaning classifiers to include automatic detection other characteristic signatures, such as: ion-acoustic, Langmuir
and solitary waves.

*Code and data availability.* The code used for this work, the trained classifiers and the training and testing data is available at: https://github.com/AndreasKvammen/ML_dust_detection. The Triggered Snapshot WaveForms (TSWF) data files can be downloaded at: https://rpw.lesia.obspm.fr/roc/data/pub/solo/rpw/data/L2/tds_wf_e/

## Appendix A:    Graphical User Interface for Manual Labeling

Figure A1 presents the Graphical User Interface (GUI) that was used to manually label all considered (3000) signals into either
*dust* or *no dust*. In addition, efforts were made to use a similar setup (with the same monitor and figure resolution) throughout
the manual labeling in order to reduce bias effects.

## Appendix B:    The Classification Performance Metrics

The classification performance metrics are calculated using the True Positive (TP), True Negative (TN), False Positive (FP)
and False Negative (FN) values, defined by comparing the predicted classes and the manually labeled classes, illustrated in
Figure 10.

The overall accuracy of the classifier is the proportion of observations that were correctly predicted by the classifier. The
accuracy is mathematically defined as:

$$\text{Accuracy} = \frac{\text{TP} + \text{TN}}{\text{TP} + \text{TN} + \text{FP} + \text{FN}} \tag{B1}$$





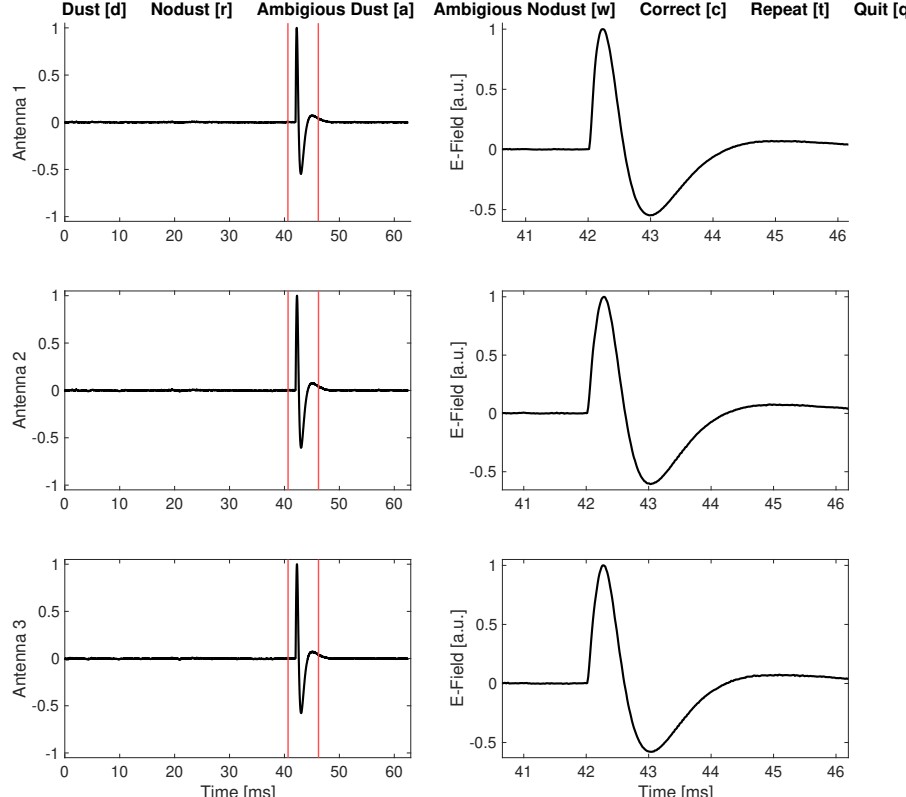

**Figure A1.** The manual labeling user interface showing a signal observed December 19, 2020. The left column displays the full snapshot (from 0 to ∼63 ms) at all antennas. An area of interest is selected by adjusting the red vertical lines. The right column displays the signal within the area of interest. The signal can be labeled as *dust* by pressing the [d] key on the keyboard and *no dust* by pressing the [r] key. The signal is indicated to be ambiguous if the waveform do not fit clearly into either of the two labels, note however that signals indicated to be ambiguous were also labeled into either *dust* or *no dust* using the [a] and [w] keys. There is also an option to correct [c] the previously labeled signal (in case of an error), repeat [t] the area of interest selection and quit [q] the manual labeling user interface.

Precision (in this case) is defined as the proportion of data points predicted by the classifier as *dust*, whose "true" label is indeed *dust*. Precision is therefore calculated as:

$$\text{Precision} = \frac{\text{TP}}{\text{TP} + \text{FP}} \tag{B2}$$

Recall (in this case) is the proportion of observations manually labeled as *dust*, that were correctly predicted as *dust* by the classifier. Recall is defined as:

$$\text{Recall} = \frac{\text{TP}}{\text{TP} + \text{FN}} \tag{B3}$$

The F1 score acts as a weighted average of precision and recall and is calculated as:

$$\text{F1} = 2 \left( \frac{\text{Precision} \cdot \text{Recall}}{\text{Precision} + \text{Recall}} \right) \tag{B4}$$



*Author contributions.* AK: Wrote the article text, trained and tested the machine learning classifiers and manually labeled the waveforms.
KW: Aided the development of the machine learning classifiers, analyzed the machine learning performance metrics and commented/edited the article. SK: Performed the dust impact rate analysis, aided with theoretical background and commented/edited the article. JV and LN: Contributed with analysis of the TDS waveforms and theoretical background. AZ and KR: Contributed with theoretical background and helpful discussions. AG: Contributed with knowledge of the Solar Orbiter data availability and discussions on the dust waveform shapes. DP and JS: Provided the data used for this work and explained the data content. IM: Is the main contributor for the theoretical background, aided the article with numerous comments/suggestions/discussions and shared knowledge that was crucial for this work.

*Competing interests.* The authors declare that there are no competing interests.

*Acknowledgements.* This work is supported by the Research Council of Norway (grant number 262941). A.K. thanks Audun Theodorsen for aiding the motivation and objective of the article. In addition, A.K thanks Juha Vierienen, Björn Gustavsson and Patrick Guio for helpful discussions. S. K. is supported by the Tromsø Research Foundation under grant 19_SG_AT. J. V., D. P. and J. S. acknowledge the support of Czech Science Foundation grant 22-10775S.



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
