# Peer review of "Machine Learning Detection of Dust Impact Signals Observed by the Solar Orbiter"

_EGUsphere, 2022_

## Author Comment (AC2)

**Reviewer 2 Comments and Response**

***Please address the capabilities of the methods in terms of the signal's lifetime and amplitudes. In case the machine learning method is replaced instead of the current onboard dust detection algorithm, does this method works for the different lifetime of the dust signals, for instance?***

We have now addressed the constraints of the method in section 5.2. Note that we refer to the "lifetime of the dust signals" more generally as the "dust impact shape" and that the amplitude is constrained by the detection threshold (~5 mV) of the RPW-TDS instrument (discussed in section 2.1 of the revised manuscript).

***Section 3.4.1 Feature Extraction: Please compare the two features selected in this study and the dust detection algorithm employed onboard TDS.***

The TDS feature extraction routine is not publicly available. We can therefore not directly compare the feature extraction techniques themselves. However, the performance of the TDS algorithm is thoroughly compared to the 2-feature SVM technique in section 4. Furthermore, a more detailed description of the TDS classifier is now included in Section 2.2.

***Figure 4: How is the 'decision line' defined?***

The decision line is defined by a polynomial of degree 2. Where the polynomial parameters are found by minimizing the non-separable SVM cost function. This is now stated explicitly in Figure 4. The mathematical formulation of the optimization problem is however not trivial and beyond the scope of this article to discuss. The curious reader is directed to Theodoridis and Konstantinos (2009) in the article.

***Figures 4 and 5: Is the similar classification confirmed for the CNN results as well?***

Yes, the same data set (the testing data) is used to obtain the CNN results. This is now stated in Figure 8.

***Figure 9: What are the highlighted area in a-i)?***

The following text is added to Figure 9 to describe the highlighted area:
"The highlighted green color indicates the CAM values associated with the *dust* class, the green regions therefore emphasize the regions that are considered important by the CNN for the *dust* class. Similarly, the red color indicates the regions that are influential for the *no dust* class."

***Figure 11: Both SVM and CNN dust detection seem to have a local minimum around the perihelion, while TDS results are largely scattered and have a maximum around the perihelion. Is there any explanation for this?***

This is an important observation that we have not noticed. The local minima may be due many reasons, now discussed in the article text. The main reason (as we propose) is that there is an asymmetry in the interstellar dust flux when going towards perihelion (upstream of the interstellar dust flux) and away from the perihelion (downstream) which might lead to a sharp

dip in the dust impact rates around perihelion. We can however not confidently state that this is the cause of the local minima around the perihelia, since the Poisson dust impact rate variation is quite large in this region, as can be seen in the updated Figure 11 (with included errorbars).

---

## Author Response (AR1)

Dear reviewers,

Thank you for reviewing our manuscript and thank you for fruitful and appropriate comments. Below is a comment-by-comment response. We have addressed all comments in the revised manuscript. However, comment 2 by reviewer 2 is not fully addressed for the reasons explained in the response below.

In addition, about 6 months of new Solar Orbiter data has been uploaded to the Solar Orbiter RPW data archive: https://rpw.lesia.obspm.fr/roc/data/pub/solo/rpw/data/L2/tds_wf_e/
This additional data is now classified and included in Figure 11.

best regards,

A. Kvammen and co-authors.

**Reviewer 1 Comments and Response**

***P2, L24-29: It may be used again but adding a reference here would be better.***

Appropriate references are added in the revised manuscript

***P2, L37, "interstellar dust population within…"?***

The word "dust" was erroneously added twice in this sentence and is now removed in the revised manuscript-

***P5, L140***

1. ***The references are given, but I still suggest the author to add a bit more details about the algorithm. For example, the range of amplitude and bandwidth seem not lengthy to be added.***
2. ***The Figure 6i event is identified as dust because the frequency is not considered in the SVM?***
3. ***Also, will Figure 1c yield a negative ratio on item 2? These seem important to help the audience to understand the performance of SVM on some not-so-typical events.***

1. A more detailed description of the TDS classification is now added to section 2.2. The amplitude threshold is also discussed in section 2.1 in the revised manuscript.

2. The Figure 6, subplot i) title had a mistype, it was titled as *dust* but should have been titled as *no dust,* this is now fixed in the revised manuscript.

3. The SVM will not have a negative convolution ratio since we only use the absolute value of the convolution. This is described in detail when we describe the SVM feature extraction routine on page 10, item 2.

***P12, L255, "Figure 6 focuses mostly" on …?***

The word "on" was missing in this sentence and is now included.

***P13, Figure 6 caption: "this can possibly be explained a weak…" ? Also, I assume that they are all 15 sec intervals, same as all such figures?***

The word "by" was missing in this sentence and is now added in the revised manuscript. We have also included text to Figures 6 and 9 to highlight that the signal framing are all 15 ms intervals.

***How many computation resources are used for the two methods? Is it trivial or expensive?***

We have included a description of the computational resources required to train the SVM classifier (in subsection 3.4.2) and the CNN classifier (in subsection 3.5.2).

In addition, we have included a discussion on the computation time needed to classify "new" observations with the SVM and CNN models at the end of subsection 4.1.

***The conclusion of the paper is that both methods work. The error improvement of CNN vs SVM presented seems trivial. In addition to the slight accuracy improvement, is there anything else to help a user choose which method to use?***

The CNN has the highest performance across all evaluation metrics. The performance advantage over the SVM/TDS classification methods is statistically significant, as shown in Table 1, we therefore suggest users to use the proposed CNN model (or a similar CNN architectures). This is now written explicitly in section 4.1. Otherwise, we can not see any significant difference between the CNN and SVM methods, both seem stable and appropriate.

**Reviewer 2 Comments and Response**

***Please address the capabilities of the methods in terms of the signal's lifetime and amplitudes. In case the machine learning method is replaced instead of the current onboard dust detection algorithm, does this method works for the different lifetime of the dust signals, for instance?***

We have now addressed the method constraints with respect to the dust impact shape in Section 4.2 (paragraph starting at line 389). We have also addressed the constraints of the method more generally in section 5.2. Note that we refer to the "lifetime of the dust signals" more generally as the "dust impact shape" and that the amplitude is constrained by the detection threshold (~5 mV) of the RPW-TDS instrument (discussed in section 2.1 of the revised manuscript).

***Section 3.4.1 Feature Extraction: Please compare the two features selected in this study and the dust detection algorithm employed onboard TDS.***

The TDS feature extraction routine is not publicly available. We can therefore not directly compare the feature extraction techniques themselves. However, the performance of the TDS

algorithm is thoroughly compared to the 2-feature SVM technique in section 4. Furthermore, a more detailed description of the TDS classifier is now included in Section 2.2 (see comment 3 by Reviewer 1).

***Figure 4: How is the 'decision line' defined?***

The decision line is defined by a polynomial of degree 2, where the polynomial parameters are found by minimizing the non-separable SVM cost function. This is now stated explicitly in Figure 4. The mathematical formulation of the optimization problem is however not trivial and beyond the scope of this article to discuss. The curious reader is directed to Theodoridis and Konstantinos (2009) in the article.

*Theodoridis, S. and Koutroumbas, K.: Chapter 3 - Linear Classifiers, in: Pattern Recognition (Fourth Edition), edited by Theodoridis, S. and Koutroumbas, K., pp. 91 – 150, Academic Press, Boston, fourth edition edn., https://doi.org/https://doi.org/10.1016/B978-1-59749-272-0.50004-9, 2009.*

***Figures 4 and 5: Is the similar classification confirmed for the CNN results as well?***

Yes, the same data set (the testing data) is used to obtain the CNN results. This is now stated in Figure 8.

***Figure 9: What are the highlighted area in a-i)?***

The following text is added to Figure 9 to describe the highlighted area:
"The highlighted green color indicates the CAM values associated with the *dust* class, the green regions therefore emphasize the regions that are considered important by the CNN for detecting dust impact signatures. Similarly, the red color indicates the regions that are influential for the *no dust* class."

***Figure 11: Both SVM and CNN dust detection seem to have a local minimum around the perihelion, while TDS results are largely scattered and have a maximum around the perihelion. Is there any explanation for this?***

This is an important observation that we have not noticed. In the revised manuscript, this observation is discussed in Sub-section 4.2 (final paragraph).